# Biomolecules of 2-Thiouracil, 4-Thiouracil and 2,4-Dithiouracil: A DFT Study of the Hydration, Molecular Docking and Effect in DNA:RNAMicrohelixes

**DOI:** 10.3390/ijms20143477

**Published:** 2019-07-15

**Authors:** M. Alcolea Palafox, A. Milton Franklin Benial, V. K. Rastogi

**Affiliations:** 1Departamento de Química-Física, Facultad de CienciasQuímicas, Universidad Complutense de Madrid, 28040 Madrid, Spain; 2PG and Research Department of Physics, N.M.S.S.V.N. College, Madurai 625019, India; 3Indian Spectroscopy Society, KC 68/1, Old Kavinagar, Ghaziabad 201002, India

**Keywords:** thiouracil compounds, hydration, molecular docking, hybrid microhelixes, DNA:RNA

## Abstract

The molecular structure of 2-thiouracil, 4-thiouracil and 2,4-dithiouracil was analyzed under the effect of the first and second hydration shell by using the B3LYP density functional (DFT) method, and the results were compared to those obtained for the uracil molecule. A slight difference in the water distribution appears in these molecules. On the hydration of these molecules several trends in bond lengths and atomic charges were established. The ring in uracil molecule appears easier to be deformed and adapted to different environments as compared to that when it is thio-substituted. Molecular docking calculations of 2-thiouracil against three different pathogens: *Bacillus subtilis*, *Escherichia coli* and *Candida albicans* were carried out. Docking calculations of 2,4-dithiouracil ligand with various targeted proteins were also performed. Different DNA: RNA hybrid microhelixes with uridine, 2-thiouridine, 4-thiouridine and 2,4-dithiouridine nucleosides were optimized in a simple model with three nucleotide base pairs. Two main types of microhelixes were analyzed in detail depending on the intramolecular H-bond of the 2′-OH group. The weaker Watson–Crick (WC) base pair formed with thio-substituted uracil than with unsubstituted ones slightly deforms the helical and backbone parameters, especially with 2,4-dithiouridine. However, the thio-substitution significantly increases the dipole moment of the A-type microhelixes, as well as the rise and propeller twist parameters.

## 1. Introduction

Pyrimidine nucleobases with sulphur appear today to have a great biological importance, since Lipsett [1] identified 4-thiouridylic acid in natural tRNAs of *Escherichia coli* in 1965. Further studies have identified other organisms as *Streptomyces libani* to produce the 4-thiouracil compound [2], or they appear as minor components in *Drosophila* tRNA (the 2-thiouridine molecule) and in prokaryotic tRNAs (the 2-thiocytidine and 4-thiouridine molecules). Thus, it is expected that thio-pyrimidines can have an effect on the arrangement of the RNA helix.

Due to the importance of thio-pyrimidines, in the present work, molecules of 2-thiouracil (2TU), 4-thiouracil (4TU) and 2,4-dithiouracil (2,4DTU) were studied. 2TU possesses several important biological properties, such as anti-carcinogenic, antifungal, antiprotozoal and antiviral activity [3,4] and it is also used as an antithyroid drug. It is also a selective inhibitor of neuronal nitricoxide synthase [5]. The chemotherapeutic activity of 2TU is due to its ready incorporation into the nucleic acid impeding the melanoma tumors growth [6]. It has been found that 2-thiouridine-enhanced the RNA hybridization [7]. 4TU also has a great medicinal importance. It is used today for labeling RNA in infected cells [8], or to analyze the RNA metabolism [9,10]. Some of its derivatives have antimicrobial properties [11].

The FTIR and Raman spectra [12,13] and the microhydration [13] of 2TU have been previously studied by us, with special attention on its effect as nucleoside on the base pairs with adenosine [13]. It has also been studied by other authors using *ab initio* [14] and density functional (DFT) methods [15,16]. Spectroscopic studies [17,18] on 4TU, as well as theoretical ones using *ab initio* and DFT methods [19,20,21] have also been reported. Its photochemistry and photobiology have been analyzed by Khvorostov et al. [22]. The molecule of 2,4DTU has been studied [23,24] much less, although we have carried out a simple spectroscopic analysis on it [25]. Water is the biological medium in which these thio-nucleobases react. However, little studies have been reported in the literature about the influence of water on the geometry, natural bond orbital (NBO) atomic charges and other properties of these thio-nucleobases [13], which is one of the goals of the present work. Molecular docking studies have also not been reported on these thiouracil nucleobases. Therefore, another goal of this work is to present the molecular docking analysis of these thiouracil nucleobases against different pathogens.

Nucleosides containing thio-nucleobases also appear as essentials and their study is an important research area in drug development. The effect of these thio-nucleosides on the geometrical parameters of the RNA strand, on the NBO atomic charges and on the Watson–Crick (WC) interactions of the base pairs is of our interest, and several mechanistic propositions have been described in the literature [7]. Thus, 2-thiouridine (s^2^U) has been found to stabilize the uracil:adenine (U:A) base pairs and to destabilize the uracil:guanine (U:G) wobble pairs [7]. 4-thiouridine (s^4^U) is also an important nucleoside but it has been little studied, and only from the thermodynamics [26] and NMR [27] point of view, together with s^2^U. On 2,4-dithiouridine (s^2,4^U) nucleoside there appears no structural and spectroscopic studies reported. Therefore, a previous work on the effect of replacement of a canonical uridine nucleoside by its analogous s^2^U, s^4^U and s^2,4^U thio-derivatives inside a microhelix was carried out by one of the authors [28]. Now the present study tries to improve this work with a better optimization of the microhelixes, and with different strands and types, which is another goal of the present study. Among the different kinds of microhelixes, the DNA:RNA hybrid appears of special interest because it is an essential biological intermediate involved in reverse transcription and DNA synthesis, and as well as in many other processes, e.g., as substrates for the enzyme RNase-H [29]. As such, these microhelixes were selected for the present study.

In resume, the present work analyzes: (i)How the distribution of water molecules around uracil, 2TU, 4TU and 2,4DTU nucleobases affect their molecular structures and charges.(ii)How the different electron distribution and charges of the thio-nucleobases under study, in special 2TU and 2,4DTU, interact with different pathogens and target proteins through a molecular docking study.(iii)How the electronegativity difference between sulphur and oxygen affects the intermolecular H-bonds between the two strands of the DNA:RNA helix, and thus in the helical parameters.(iv)How the positions of sulphur and the nucleosides in the strand affect the helical parameters in three different types of optimized microhelixes.

## 2. Computational Methodology

All the density functional (DFT) calculations were carried out using the B3LYP and the Minnesota functional M062X [30] exchange–correlation functionals and with the 6-31G(d,p) basis set through the Gaussian09 program package [31] running in the Computational Center of the Complutense University of Madrid. Standard parameters have been used under the UNIX version of the program. The B3LYP method was selected for the simulation of all the hydrated structures of the thio-nucleobases because it is one of the most widely used method today, and it has been developed most adequately in many spectroscopic studies [32,33], as well as in the analysis of nucleosides [34,35,36], DNA base pairs [37], hydrated nucleobases [38], in drug designing [39], etc. The M062X exchange–correlation functional was selected for the simulation of all the DNA:RNA microhelixes because previous calculations [13,28] with several DFT methods show that it determines values in accordance to the experimental, while B3LYP give rises to non-parallel base-pair planes in the helix in contrast to the experimental. The Berny algorithm was used for the optimization process under the standard convergence criteria. The default fine integration grid was utilized. Frequency calculations were also performed to assess that all the geometries obtained really correspond to stationary points and minima.

The Modified Scheme of Monosolvation (MSM) [40] was used for the hydration of all the thio-nucleobases up to 30 water molecules, which includes the first and second hydration shells. This methodology has been utilized most successfully by us earlier in the simulation of the first hydration shell of 2TU [13], and in several nucleobases [41,42] and nucleosides [43,44].

The molecular docking analysis was performed using iGEMDock [45,46] and AutoDock Tools-1.5.4 interfaced with the MGL Tools-1.5.4 package [47] and the ligand-protein binding pose was visualized by PyMOL molecular graphics system (version 1.7.4.5 Edu) [48]. The ligand PDB file was generated by using the optimized molecular structure of the molecules. In order to select the starting structure of the ligand molecules, the preliminary molecular docking computations were performed to remove clashes among atoms of the ligand and to develop a reasonable starting pose using the Avogadro software (version 1.2.0). The three-dimensional (3D) coordinates of the targeted proteins were downloaded from the Research Collaboratory for Structural Bioinformatics (RCSB) protein data bank [49,50], with a resolution of 2.5 Å. The targeted proteins were prepared by the following steps:(i)The AutoDock Tools graphical user interface [47] was used to remove the ligand and water molecules present in the targeted proteins and,(ii)Then polar hydrogen and Kollman charges were added in the targeted proteins.(iii)The AutoGrid 4.2 [47] was used to create affinity grid centered on the active site with grid size 126 × 126 × 126 with a spacing of 1.0 Å. The rigid targeted protein and flexible ligand docking were performed by using AutoDock 4.2 with the Lamarckian genetic algorithm applying the following protocol: Trials of 100 dockings, energy evaluations of 25,000,000, population size of 200, a mutation rate of 0.02, a crossover rate of 0.8 and an elitism value of 1.

Three types of DNA:RNA microhelixes with thiouridine molecules were only optimized and considered here. They were simulated first with a global negative charge of −4. Only in few of these microhelixes were additionally included one Na atom on each phosphate moiety. In total four Na atoms were in the microhelix, and in this case the charge was changed to 0. To guarantee that both strands of the helix correspond to the same type, a previous simulation with the corresponding RNA:RNA helixes was carried out. Further removing of the 2′-OH groups of one strand and a full optimization process leads to the DNA:RNA microhelixes studied here. This consideration improves the previous results published on them by one of us [28], and gives rise to noticeably more stable helixes (higher negative energy). The hydration of several selected of these microhelixes was also performed using distinct number of water molecules surrounding the skeletal backbone, and in general distributed around the sodium atoms and the phosphate groups.

## 3. Results and Discussion

### 3.1. Geometry Optimization and Atomic Charges in the Isolated State

The molecules of uracil (U), 2TU, 4TU and 2,4DTU nucleobases were fully optimized using different levels by MP2 and DFT methods. A brief resume of selected geometric parameters obtained is collected in Table 1, while the labeling of the atoms appears plotted in Figure 1. Since only experimental values have been reported for uracil and 2TU molecules, therefore for comparison purposes the X-ray values for 2TU have also been included in Table 1. The molecule of 2TU appears in the crystal as full planar and H-bonded through the sulphur and oxygen atoms with the amino hydrogen atoms of the neighboring molecules in a linear polymeric arrangement [51]. In general, the calculated values of geometric parameters appeared close to the experimental ones. The optimized C=S bond length of 1.665 Å by B3LYP in 2TU was very close to that of 1.663 Ǻ in 4TU, and to 1.662 and 1.660 Ǻ in 2,4DTU. These values were slightly larger than those reported for the C=S double bond (1.61 Å), and too short for the single bond [12], 1.79 Å. They were also shorter than that reported in the crystal of 2TU by X-ray, 1.683 Å, because of the arrangement of H-bonds involving the sulphur atom [51].

The substitution of an oxygen atom by a sulphur atom on the uracil ring leads to a significant change in the bond-length at the site of substitution: S=C ~1.65 Å, as compared to C=O ~1.23 Å. This fact leads to a slight shortening in the neighboring bond lengths and a closing of the related *ipso* angle. Thus, it is apparent that when the oxygen atom at 2nd position in uracil is replaced by the sulphur atom, the N1–C2 and N3–C2 bonds are shortened by ca. 0.015 Ǻ and the N1–C2–N3 angle is slightly opened, ca. 1°. Analogously, when the oxygen atom at position 4 is replaced by the sulphur atom, the N3–C4 and C4–C5 bonds lengths are shortened by ca. 0.02 Ǻ and an opening of the N3–C4–C5 angle of ca. 1° is also noted, Table 1. This lengthening of the bond lengths in the uracil ring when there is a thio-substitution, leads to a slight increment in the flexibility of the ring, and a decrease in the contribution of the resonant form into the structure [52]. The C–H and N–H bond lengths of the uracil ring, as well as the C5=C6 bonds, appear little affected by the thio-substitution on O2 and/or O4, in accordance to Singh et al. [23].

The values of the dipole moment (μ) in thiobases were slightly larger than in the uracil molecule. Thus for example, in uracil it was 4.248 D (Table 1) while in 2TU was 4.621 D, in 4TU was 5.064 D and in 2,4DTU was 5.076 D. Although the thiobases had larger values of μ than the canonical forms with oxygen, the main direction of μ remained the same in both, thiobases and canonical forms.

The calculated natural atomic charges (NBO) by two theoretical methods are shown in Table 2. As apparent from Table 2 in general, the results by the DFT values were similar for uracil, 2TU, 4TU and 2,4DTU molecules, with the exception of those around the sulphur atoms. The values by MP2 differ largely from those by B3LYP. The negative charge on the sulphur atoms appeared about three times lower than that on the oxygen atoms. With the sulphur substitution it led to an increment in the negative charge on the neighboring nitrogen atoms, and to a slight increment in the positive charge on the bonded hydrogen atoms H7 and H9, ca. 0.01*e* (where *e* is the charge of an electron), as compared to uracil molecule. It also led to a huge reduction in the positive charge on the C2 and C4 atoms, the highest ones in the uracil molecule.

### 3.2. Hydration of the Nucleobases

One of the goals of the present study was to know how the water distribution in 2TU, 4TU and 2,4DTU thionucleobases was different, and how this water distribution affected their molecular structures. This influence was studied with a variable number of explicit water molecules surrounding 2TU, 4TU and 2,4DTU.

Following the MSM scheme, Appendix A depicts as examples the optimum hydrated structures of 2TU with two to fifteen water molecules. Two views (general and lateral) are shown. The corresponding total energy is shown in the bottom of each figure. The water molecules were numbered as w*_i_* (*i* = 1 to *n*), where w_1_ indicates that this water molecule was the strongest H-bonded to 2TU in the 2TU (H_2_O) hydrate. Similarly, w_2_ corresponds to the second strongest H-bonded water molecule to 2TU in 2TU (H_2_O)_2_, and so on. For simplicity, Figure 2 shows the optimized structure of 2TU with 20 water molecules, which includes the first hydration shell and a part of the second hydration shell, while Table 1 collects values of the main geometrical parameters in the clusters with 30 water molecules. The values of the NBO atomic charges in these clusters with 30 water molecules are included in Table 2.

Due to the lower attraction of sulphur, the water molecules are stronger H-bonded to the oxygen atoms than to the sulphur atoms. It leads to: (i)A slight difference in the distribution of water molecules around the sulphur atoms as compared to that around the oxygen atoms, as indicated in a circle in orange color in Figure 2.(ii)Values of the deformation energies E^def^ and interaction energies are higher in the cluster with uracil U(H_2_O)_20_ than in the related clusters with 2TU [13], 4TU and 2,4DTU molecules.

The thiouracils under study appeared as flexible molecules, with a full planar geometry in the isolated state, and with a slight bending in the first and second hydration shells. The water molecules slightly affected the planarity of the uracil ring, in general with changes lower than 7° in their torsional angles, i.e., the value of the C6-N1–C2–N3 torsional angle was −6.3° in U, and −6.6° in 4TU, Table 1.

A high conformational flexibility of the pyrimidine ring represents one of the possible ways of geometry relaxation for the creation of the most favorable conditions for inter-and intra-molecular interactions with other molecules [53]. 

The following features were observed with the progress of the hydration of uracil, 2TU, 4TU and 2,4DTU molecules:(i)In the isolated state 2TU was slightly less stable than 4TU. However, the hydration changed this stability and with several water molecules 2TU appeared more stable than 4TU.(ii)The trend of shift in different parameters of these nucleobases with the progress of the hydration appeared in general similar, as shown in Figure 3, Figure 4 and Appendix A.(iii)A strengthening of the intermolecular H-bonds of the nucleobases appears with the hydration with a noticeable increment of this strengthening in the formation of the first hydration shell with about 10 water molecules, and a slight further increment in the second hydration shell with about 30 water molecules, Figure 3. However, as an exception the intermolecular H-bond H3(N3)···O_w_ was found stronger in the 1st hydration shell than in the 2nd hydration shell. It is also noted the large lengthening in the H3 (N3)···O_w_ H-bond and the small lengthening in O2···H_w_, O4···H_w_ and S4···H_w_ H-bonds after the 1st hydration shell was raised. It seems that after the 1st hydration shell, the number of water···water interactions increase over the nucleobase···water, and the interactions of the new water molecules with the water molecules directly H-bonded to the nucleobase leads to a slight weakening of the nucleobase···water H-bonds, because the water···water interactions are remarkably stronger than nucleobase···water.

With the progress of the hydration and the formation of water net up and down of the nucleobase plane, the H-bonds of the water molecules compress the structure, approaching the water molecules nearer to the nucleobase, which leads to an increment in the nucleobase···water interactions.
(iv)The H1(N1)···O_w_ H-bond appeared slightly stronger than H3(N3)···O_w_. With the progress of the hydration, the shortening in the H1(N1)···O_w_ H-bond was also slightly larger than in H3(N3)···O_w_.(v)The shortening of the S···O_w_ H-bonds appeared higher than O···O_w_ H-bonds. Thus for example, this shortening in 2TU and 4TUmolecules as compared to uracil was 0.160 Å with O2 and 0.163 Å with O4, while in 2,4DTU was 0.255 Å with S2 and 0.251 Å with S4.(vi)The C5-H···O_w_ and C6-H···O_w_ H-bonds of hydrophobic sites were very weak in the beginning of the hydration. However, with the progress of the hydration and the formation of a water net that compressed the structure, these H-bonds appeared with values around 2.0–2.3 Å, and thus became prominent. The C6-H···O_w_ H-bond appeared slightly stronger than C5-H···O_w_ in all the clusters, e.g., in the cluster U(H_2_O)_20_ C5-H···O_w_ = 2.307 Å and C6-H···O_w_ = 2.292 Å, while in U(H_2_O)_30_ the values were 2.365 and 2.184 Å, respectively. In 2TU, 4TU and 2,4DTU these H-bonds appeared slightly shorter, e.g., in 2,4DTU(H_2_O)_20_ the value of C5-H···O_w_ = 2.300 Å and C6-H···O_w_ = 2.013 Å, while in 2,4DTU(H_2_O)_30_ the values were 2.236 and 2.203 Å, respectively.(vii)On hydration, the molecular structure of the nucleobase was changed noticeably with longer C=O/C=S bond lengths and shorter adjacent N1–C2 and C2–N3 bond lengths, Figure 4, in addition to other transformations in the structure of the nucleobase. The largest change in the bond lengths appeared in the formation of the 1st hydration shell with a further slight and prolonged lengthening/shortening of the bond length with the progress of the hydration. Similar features have been observed by us in the hydration of other related nucleobases [41,42]. It was noted a similar trend with the hydration of the C=O and C=S bonds, Figure 4.(viii)The C2–N3 bond length changed in an irregular way with the hydration, with a shortening in the 1st hydration shell with 8–10 water molecules and then a lengthening of C2–N3 bond up to 20 water molecules. In the formation of the 2nd hydration shell with up to 30 water molecules a new and noticeable shortening of C2–N3 bond appeared again.(ix)An opening of the *ipso* angles on C2 and C4 (corresponding to the N1–C2–N3 and N3–C4–C5 angles), and related closing of the neighboring angles C6-N1–C2 and N3–C4–C5 was observed with the hydration. For example, in the uracil molecule, an opening of 4° in angle N1–C2–N3 and 3° in angle N3–C4–C5 was calculated from the isolated state to the cluster with 30 water molecules, and in contrast a closing of 2° in angle C6-N1–C2 and 4° in angle C2–N3–C4. However, with the hydration slightly lower values were found in 2TU, 4TU and 2,4DTU molecules, e.g., in 2,4DTU was found an opening of 3.4° in N1–C2–N3 and 2.3° in N3–C4–C5, and a closing of 1.4° in C6-N1–C2 and 3.4° in C2–N3–C4.(x)The uracil ring was full planar in the isolated state of all the molecules under study. At the beginning of the hydration this planarity remained maintained. However, in the first hydration shell this planarity of ring disappeared, and becomes out-of-plane around 1–2°, and this deformation remarkably increased with further hydration. The maximum deformation appeared in the clusters with about 15–20 water molecules, with changes around 7–10°. Thus, in U (H_2_O)_20_ the N1–C2–N3–C4 torsional angle had the large value of −8.1°, the torsional angle C2–N3–C4–C5 had 11.4° and N3–C4–C5-C6 had −9.9°. Since the attraction between the water molecules and the sulphur atoms was lower than the attraction between the water molecules and the oxygen atoms, the ring deformation was noticeably less in the thiouracil molecules as compared to uracil. Thus, for example in 2,4DTU (H_2_O)_20_ the torsional angles N1–C2–N3–C4, C2–N3–C4–C5 and N3–C4–C5-C6 had the values 0.2°, 5.7° and −8.6°, respectively.

With further hydration and on the formation of the 2nd hydration shell the water molecules compress the structure of the nucleobase and the out-of-planarity was remarkably reduced, e.g., in U(H_2_O)_30_ the angle N1–C2–N3–C4 = 5.4°, the angle C2–N3–C4–C5 = −0.4° and N3–C4–C5-C6 = −3.9°, while in 2,4DTU(H_2_O)_30_ the values of these angles were 3.0°, 1.4° and −3.9°, respectively.

The ring in uracil molecule appeared more deformed than in its thio-substituted structures. This high flexibility of the ring in uracil molecule could be related to its great biological importance, in special for the genetic code. By contrast, a lower flexibility of the uracil ring with thio-substitution could explain the less biological importance of these thio-molecules.
(xi)The uracil molecule had the highest negative charges on N1, N3, O2 and O4 atoms, as compared to in 2TU, 4TU and 2,4DTU molecules, Table 2 and Appendix A. This feature shows that in uracil molecule the intermolecular H-bonds through these atoms (N1H, N3H, O2, O4) with the water molecules were stronger than in 2TU, 4TU and 2,4DTU molecules. Moreover, the lower attraction of the thiouracil molecules with water could facilitate the removing of these water molecules when in the primer of the helix begin the establishment of the WC pairs. The lower attraction of the thiouracil molecules with other molecules, such as adenine could also deform the growing of the RNA helix, as explained in Section 3.4.(xii)The sulphur atom in S2 position remarkably reduced the negative charge on N1, ca. 0.06*e*, but this reduction on N3 atom was small when the sulphur atom was in position S4, as compared to the uracil molecule. Thus, the NBO negative charge on N1 atom had the lowest value in the 2,4DTU molecule. This negative charge on N1 was also reduced with the hydration.(xiii)The negative charge on the N3 atom was reduced in a similar amount, ca. 0.06*e*, by the sulphur atoms in S2 and S4 positions. Thus, in 2,4DTU molecule this charge was reduced to two times and therefore N3 atom appeared with the lowest charge and close to that on N1.(xiv)The charge on O2/S2 atom was little affected by the substitution of O4 by S4 atom or vice versa. Similarly, the charge on O4/S4 was also little affected by the substitution of O2 by S2 atom or vice versa. Since the negative charge on the nitrogen atoms was reduced with the hydration, the negative charge on the oxygen atoms was increased by a similar amount with the hydration. The trend of this increase in negative charge was similar in uracil, 2TU, 4TU and 2,4DTU molecules.(xv)The dipole moment (µ) was remarkably changed depending on the spatial distribution of the water molecules. Due to the large polarity of the water molecules, small variations in their distribution can create large differences in µ. With the progress of the hydration an irregular trend of change in the values of μ was observed in all the molecules under study, Appendix A. The highest value appeared in the cluster with 20 water molecules.

### 3.3. Molecular Docking Analysis

Molecular docking has become an increasingly important tool for drug discovery. The molecular docking approach can be used to model the interaction between a small molecule and a protein at the atomic level, which allows us to characterize the behavior of small molecules in the binding site of target proteins as well as to elucidate fundamental biochemical processes [54]. Generally, the thiourea-based compounds such as 6-n-propyl-2-thiouracil, MMI, 6-methyl-2-thiouracil and carbimazole are widely used as anti-thyroid drugs [55]. In the present study, the molecular docking analysis was carried out for 2TU against three different pathogens, and for flexible ligand of 2,4DTU and the rigid targeted proteins TSHR (PDB ID: 2XWT), TPO (PDB ID:1VGE) and NIS (PDB ID:1PW4). Moreover, the molecular docking analysis was performed for MMI ligand and NIS (PDB ID: 1PW4) targeted protein. MMI is a well-known anti-thyroid agent. The ligand molecules were docked well with all the targeted proteins.

#### 3.3.1. Molecular Docking of 2TU

The in silico biological screening of 2TU against three different pathogens such as, *Bacillus subtilis* (G^+ve^), *Escherichia coli* (G^−ve^) and *Candida albicans* were carried out. The respective protein data bank codes are 4OZ5, 5C59 and 2Y7I. Various intermolecular N-H···O, C-H···O, O-H···N, S-H···O and C-O···N interactions explained the efficiency of 2TU on the selected microorganisms. Molecular docking of 2TU against 4OZ5 revealed a strong intermolecular H-bonded interaction, which was observed between the N–H site of ASN33 and O atom of 2TU (N-H···O, 2.03 Å), and with the N–H site of GLU32 and O atom of 2TU (N-H···O, 2.22 Å). The total binding energy of the system was calculated as −59.59 kcal/mol. Similarly for the gram negative bacteria (*Escherichia coli*—5C59) an enormous number of H-bonds were observed.

A strong H-bond was observed between the O–H site of TYR84 and O atom of 2TU with a distance of 2.34 Å. Thio group of 2TU on interaction with O atom of GLY21 site (2.25 Å) brought about the evidence of the strong binding nature of the ligand. *Candida albicans*, a fungal strain, also exhibited a great inhibition against 2TU. Strong intermolecular interactions between the N–H site of GLY25 and O atom of 2TU (3.07 Å) and between the N–H site of GLY27 and O atom of 2TU (3.099 Å) were also observed. A strong C–O···N intermolecular H-bond was observed between THR28 (C-O) and 2TU (N), 2.59Å.

Table 3 and Figure 5 explain the binding energy profile and interactions of 2TU against 4OZ5, 5C59 and 2Y7I.2TU had a greater binding energy with *Escherichia coli*, a gram negative bacteria, than with a gram positive and fungal strain. From the in silico results, it was concluded that 2TU could be more effective on gram negative strains.

#### 3.3.2. Molecular Docking of 2,4DTU

The docking results of 2,4DTU were evaluated by sorting out the binding free energy predicted by docking confirmations. The lowest energy docked poses of the ligand with various targeted proteins are shown in Figure 6, which shows the preferred binding orientation of the ligand molecule. The dotted yellow lines show the hydrogen bond formation between the ligand and targeted proteins. The lengths of the hydrogen bonds along with the amino acid in the targeted proteins are shown in Figure 6. The docking parameters for the first three ranks (1, 2 and 3) such as binding energy, inhibition constant and intermolecular energy of the ligand with the targeted proteins are listed in Table 4. These results indicate that the ligand (2,4DTU) exhibited the lower binding energy and inhibition constant for the targeted protein NIS. Hence, all these results will be useful for the development of anti-thyroid drugs.

##### Target Proteins Studied

Thyroid gland is comprised of spherical follicles filled with colloidal shape cuboidal epithelial cells called as follicular cells, which are responsible for iodine uptake and thyroid hormone synthesis [56]. The changes in the follicular cells size and shape cause the too much (called as hyperthyroidism) or too little (called as hypothyroidism) secretion of thyroid hormone, which leads to thyroid cancer. Thyroid cancer occurs more commonly in women over 50 years of age [57]. Several key protein molecules including, thyroid-stimulating hormone receptor (TSHR), thyroid peroxidase (TPO) and sodium iodide symporter (NIS), are responsible for thyroid cancer [58,59].

TSHR is a main protein in the regulation of thyroid function, which causes the gland to make and release thyroid hormone into the blood. Human TSHR (hTSHR) is a protein to be the foremost autoantigen in autoimmune hyperthyroidism in case of thyroid cancer [60]. The small molecule inhibitors can stop the actions of TSHR-stimulating immunoglobulins, and therefore can be useful in the drug development for thyroid cancer [61].

TPO (thyroid peroxidase) is an important enzyme for thyroid hormone biosynthesis and a major source for an autoantigen in thyroid cancer [62]. In general, the presence of high levels of antibodies in TPO is useful in the diagnosis of the disease. TPO is useful in iodination of tyrosyl residues in thyroglobulin and other proteins related to thyroid cancer. The inhibition of TPO by propylthiouracil (PTU) and methimazole (MMI) molecules was reported [63] and hence TPO is a specific target for thyroid cancer drug development.

NIS is a transmembrane glycoprotein that participates in the regulation of Na^+^/I^−^ gradient and transports two sodium ions (Na^+^) for each iodide ion (I^−^) into the cell. NIS mediated uptake of iodide ion into follicular cells of the thyroid gland is the initial stage for thyroid hormone biosynthesis [64]. Mutations in the NIS DNA lead to hypothyroidism and thyroid dyshormonogenesis [65]. Hence the NIS is a specific target for the thyroid cancer drug development.

### 3.4. WC Base Pairs with the Nucleosides 2-Thiouridine, 4-Thiouridine and 2,4-Dithiouridine

In the conformational analysis of the nucleoside uridine (U) in the isolated state by the M062X exchange–correlation functional we observed that the furanose ring appears in the C3′-*endo* form (^3^E symmetry) in the global minimum, although with a difference [28] of only 2.58 kJ·mol^−1^ with the C2′-*endo* (^2^E symmetry). This small difference has been confirmed by a NMR study in solution in which both the ^2^E and ^3^E forms appear in rapid equilibrium [51]. The barrier height of this equilibrium has been reported to be slightly larger with the s^2^U nucleoside than with U [7].

The influence of the sulphur atom was also analyzed when it was inserted in the base pairs with dA. Therefore, the WC base pairs uridine:adenosine (U:dA), 2-thiouridine:adenosine (s^2^U:dA), 4-thiouridine:adenosine (s^4^U:dA) and 2,4-dithiouridine:adenosine (s^2,4^U:dA) with nucleosides were optimized. The notation dA corresponds to 2′-deoxyadenosine molecule, and dG to 2′-deoxyguanosine. In these WC base pairs, one of the nucleosides (U, s^2^U, s^4^U, s^2,4^U) has a ribose ring, and another a 2′-deoxyribose ring (in dA). In the WC U:dA pair, we have observed that in the global minimum, the uridine nucleoside also appears in the C_3_-*endo* form. The sugar conformation is of great interest because it establishes the shape of the helix.

The s^2^U:dA base pair has been reported to have a larger thermodynamic stability than the U:dA base pair, which suggests that substituting uridine (U) nucleoside by s^2^U nucleoside might increment the fidelity and rate of the non-enzymatic copying of RNA templates [66,67]. The sulphur atom also affects the calculated entropy of its WC pair. Thus, with s^2^U it is slightly higher (904.1 J/mol·K) than that with uridine (897.0 J/mol·K), which could be due to its weaker intermolecular H-bonds.

#### Effects of the Sulphur Atom on the Geometrical Parameters of the WC Pair

Another goal of the present work was to identify the influence of sulphur on the geometrical parameters, NBO atomic charges and intermolecular H-bonds of its WC pair, as compared to the related canonical U:dA pair. For this task, a graphical comparison of the main effects of s^2^U, s^4^U and s^2,4^U on the base pair is resumed in Figure 7, in which a few interatomic distances and charges were included, whereas the main intermolecular angles are shown in Table 5. The most important characteristic parameters determined in the WC pairs with nucleobases, nucleosides and nucleotides are summarized in Table 6. As it was observed, the sulphur atom in position 2 (s^2^) was not H-bonded with dA. However, it had influence on the values of the calculated interaction energies and charge distribution of the WC pair, which appeared slightly different with that of uridine. This feature with s^2^U gave rise to a slight opening of the base pair angle N3-H···N^A^ toward the minor groove side, Figure 7. With s^4^U the effect was the reverse, with a closing of this angle. Another effect of the thiobases was that they could also somewhat alter the geometrical structure of the DNA:RNA helix, stabilizing the base pairs with dA vs. those with uridine, and destabilizing the wobble pairs with dG {26].

The intermolecular H-bonds C4=O···H6^A^-N6^A^ and N3-H···N1^A^of the U···2′-dA base pair (see Figure 7 for notation) were remarkably affected by the sulphur substitution, because:

(i) A C=S bond was noticeably longer than the C=O bond. It was 0.448 Ǻ in 2TU (0.426 Å by MP2, Table 1) and 0.444 Ǻ in 4TU nucleobases, and with similar values in their corresponding nucleosides, s^2^U and s^4^U. Therefore, the S4···H^A^ H-bond distance appeared longer than the O4···H^A^ distance, in the nucleobases as well as in the nucleoside base pairs, i.e., 0.520 Ǻ with s^4^U. It could be explained by a lower electronegativity of the sulphur than the oxygen atoms, and thus their weaker H-bond acceptor abilities than the oxygen atoms.

(ii) In the base pair formation, the lengthening of the N3–H and N6^A^–H^A^ bonds was reduced with a thio-uracil derivative. Thus, with s^2^U nucleoside, N3-H···N1^A^ H-bond appeared weaker and the O4···H^A^-N6^A^ stronger than that with corresponding bonds with uridine. Although, the reinforcement in the O4···H^A^–N^A^ H-bond was ca. half that the weakening of the N3–H···N^A^ H-bond. In the pair with s^4^U nucleoside, as well as with s^2,4^U, both H-bonds were weakened noticeably. With the s^4^U nucleoside a new and weak H-bond/interaction O2···H2^A^–C2^A^ appeared in the base pair, which slightly stabilized the base pair but lower than the destabilization by the sulphur atom, Figure 7.

### 3.5. MicrohelixesDNA:RNA with the Sulphur Atom

The accuracy of three DFT methods in the simulation of a DNA:RNA microhelix was firstly tested using only two nucleotide base pairs, Figure 8, Appendix A. The selected DFT methods were B3LYP, M052X and M062X, and the simulated microhelixes were 5′-s^2^U U-3′, 5′-s^2^U C-3′ and 5′-s^2^U s^2^U-3′. The comparison was mainly focused on the obtained intermolecular H-bond values. In general, similar values of these H-bonds were determined by M052X and M062X exchange–correlation functionals, while those by the B3LYP method differed remarkably. In addition, because the B3LYP method failed in the stabilization of the helix by staking interactions of the base pair planes [28], therefore it was not considered in further calculations. The calculations also failed using the LC-wPBE/6-31G(d,p) level, Appendix A. By the M062X exchange–correlation functional these interactions were slightly better simulated than by M052X, and thus it was the method selected for all the computations in the microhelixes with three nucleotide base pairs.

The intermolecular H-bonds C4=O···H6^A^–N6^A^ and N3–H···N1^A^of the U···2′-dA base pair (see Figure 7 for notation) were remarkably affected by the sulphur substitution, because:

A DNA:RNA hybrid helix was formed by two strands. For convenience, strand *I* corresponded to RNA while strand *II* to DNA. Thus, in RNA the sugar ring is a β-d-ribose, while in DNA it is a β-d-2′-deoxyribose. With this micro-simulation with three nucleotide base pairs, the optimized parameters of the nucleotide in the plane *n* (Scheme 1) can be something close to those that could be found in a long helix. Thus, only the values obtained through these nucleotides were included in the Table 6 and Appendix A of the present manuscript.

The labeling of the atoms in the microhelix appears plotted in Scheme 1, and its numbering correspond to that following the Saenger’s standard notation [68]. This notation was also used for the description of the exocyclic (χ, ζ, α, β, γ, δ, ε, ε′) and endocyclic (ν_0_ to ν_4_) torsional angles in the nucleotides of the helix. The subscripts (0, 1 or −1) refer to the nucleotide pair plane (n, n + 1 or n − 1, respectively), and the superindex A to 2′-deoxyadenosine (dA). The values of the most characteristic geometric parameters of the microhelixes are shown in Table 6 and Appendix A. They were mainly obtained in strand *I* and the notation used only represented this strand. In this way, the helix: 
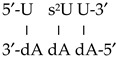
 with the two strands was represented by only one of the strands (the RNA), such as this notation: 5′-U s^2^U U-3′. In this way the helix: 
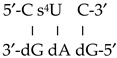
 was represented by this notation: 5′-C s^4^U C-3′, and the helix: 
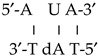
 (where T = thymidine) by 5′-A U A-3′, and so on with the remaining structures included in these Table 6 and Appendix A, i.e., in the notation used for the microhelixes the RNA strand was only represented.

#### 3.5.1. Different Inter- and Intra-Molecular H-Bonds Found in the Simulated Microhelixes

Fourteen distinct H-bonds have been observed in the simulated A-, B-and C-type microhelixes. We have labeled them as ① to ⑭, according to a previous publication [28]. Four corresponds to intermolecular H-bonds between strands *I* and *II* (①, ②, ⑨ and ⑫, and ten to intramolecular (③ to ⑬) in the same strand. Scheme 2 shows several of them in the microhelix 5′-A U A-3′, while several characteristic H-bond length values appear shown in Appendix A. The description of them is as follows:
(i)With uridine/s^2^U/s^4^U/s^2,4^U and adenosine H-bond ① corresponds to N3-H3···N1^A^ and H-bond ② to O4···H6^A^-N6^A^. Both H-bonds were the most important and responsible for the double helix maintenance. H-bond ① was found slightly longer than H-bond ②. With the cytidine (C) and guanosine (G) nucleosides appeared a new H-bond ⑫ (not included in Scheme 2), N4^C^–H4^C^···O6^G^ between NH_2_ of C and O6 oxygen of G.(ii)H-bond ③ refers to O5′_−1_–H5′_−1_···O4′_−1_ in the plane n − 1, and it appeared by rotation of the O5′H group to the most stable orientation β_+_. This H-bond will not be observed in the helix.(iii)H-bonds ④, ⑧ and ⑭ were responsible for the formation of the A-, B-and C-types microhelixes, respectively. These are very important H-bonds and appear as a consequence of the high reactivity of the H2′_n_ hydroxyl hydrogen bonded to O4′_n+1_ (A-type), to O7’_n+1_ (B-type), or to O5′_n+1_ (C-type). Of these H-bonds, the strongest one corresponds to H-bond ⑧.(iv)H-bond ⑤ refers to O2′_1_–H2′_1_···O3′_1_ of the primer (plane n + 1) and it appeared only on the RNA strand.(v)H-bond ⑥ corresponded to C6–H6···O5′ in pyrimidines and C8^A^–H8^A^···O5′^A^ in purines. It is a weak H-bond/contact that was found only in A-type microhelixes, which increased the stabilization of this microhelix together with H-bond ④.(vi)The weak H-bond ⑦ referred to C$8_{-1}^{A}$–H$8_{-1}^{A}$···O$7\prime_{-1}^{A}$, and it was found only in the plane n − 1 and in the 5′-UUU-3′ microhelix of A-type (not included in Scheme 2). This H-bond would not be observed with the rising of the microhelix.(vii)H-bond ⑩ corresponded to N$6_{1}^{A}$-H$6_{1}^{A}$···O$4_{0}^{}$ between the NH_2_ amino hydrogen of adenosine in plane n + 1 and the oxygen atom of uridine in plane n. It appeared only in the 5′-AUA-3′ microhelix of B-type.(viii)The weak H-bond ⑪ referred to C$8_{1}^{A}$-H$8_{1}^{A}$···O$2\prime_{0}^{}$ between the hydrogen H8 of adenosine of the plane n + 1 and O2′ of the ribose ring of plane n. It is found in the RNA strand of 5′-AUA-3′microhelixes of B-type.(ix)The intra-strand H-bond ⑬ corresponds to N$4_{-1}^{C}$-H$4_{-1}^{C}$···N$6_{0}^{A}$ between the NH_2_ amino group of cytosine (plane n − 1) and adenine (plane n; not included in Scheme 2), which was found in 5′-CUC-3′microhelixes. 

#### 3.5.2. Types of DNA:RNA Hybrid Microhelixes Optimized 

Four types of microhelixes were simulated depending on the H-bonding of the hydroxyl hydrogen H2′(O2′) of the RNA strand, which could be H-bond to O4′ of the next nucleotide (A-type), to O7′ of the phosphate moiety (B-type), Scheme 3, to O5′ of the adjacent nucleotide (C-type) or to O3′ of the same furanose ring (D-type), Appendix A. The notation used for the A-type and B-type microhelixes [28] appeared in agreement with the structural characteristics of double helixes [68], in which A-type corresponds to C3′-*endo* (^3^E symmetry) and B-type to C2′-*endo* (^2^E). The notations C-type and D-type are for our convenience. These microhelixes can be described as follows:

(i)A-type: Corresponds to the O2′–H2′_(n)_···O4′_(n+1)_ H-bond ④ between contiguous nucleotides of the same strand, Scheme 3a,c, and with the pyrimidine ring in high-*anti* orientation relative to the furanose ring.(ii)B-type: Corresponds to the O2′–H2′_(n)_···O7′_(n+1)_ H-bond ⑧ between contiguous nucleotides of the same strand, Scheme 3b,d, and with the pyrimidine ring appearing in *anti* orientation related to the furanose ring. These microhelixes always appear remarkably more stable than A-type microhelix, mainly because H-bond ⑧ is noticeably stronger than H-bond ④.(iii)C-type: Corresponds to the intra-strand O2′–H2′_(n)_···O5′_(n+1)_ H-bond ⑬ between adjacent nucleotides, Appendix A. This type of microhelix was only optimized in 5′-AUA-3′, and because it was significantly less stable than both A-type and B-type microhelixes, therefore its values were omitted in the present manuscript.(iv)D-type: Corresponds to the O2′–H2′_(n)_···O3′_(n)_ H-bond/interaction in the same nucleotide. This type of microhelix was found only stable when it was mixed with C-type in 5′-AUA-3′, Appendix A. Because it was the least stable type of the simulated microhelixes its values were also omitted in the present manuscript.

The A-type microhelix is characterized by a wider helix (diameter *d*, Appendix A in which the base pairs are pushed outwards from the helix axis resulting in a hole down the middle of the helix. This microhelix appears with a dipole moment (μ) lower than B-type, which indicates that it is less stable in water solution, similar to that observed by us in the 5′-UUU-3′ and 5′-AUA-3′RNA:RNA microhelixes. Therefore, because of its lower stability, it is expected that A-type microhelix will not appear in the DNA:RNA helixes, and in contrast, B-type helix could be the feasible form. However, A-type are the forms that appear in the RNA duplex [52,69]. It can be explained as follows: As a nucleoside form, the C3′-*endo* form ^3^E (A-type) of uridine is slightly more stable than ^2^E (B-type), as we have determined in the isolated state as well as under the hydration form, i.e., in the primer, the most stable form of the ribose ring is in the ^3^E form. Moreover, in this form the 3’-OH group is in the line of nucleophilic attack by the next nucleotide [69]. Thus, the helix will grow in the ^3^E form. ^2^E form can appear only when the 2′-OH group is H-bonded to O7′ of phosphate, but it does not occur in the primer, in which in the ^3^E form the 3’-OH group has the most appropriate orientation to bond to the next nucleotide. Thus, the B-type helix with the ^2^E form will be only possible through an internal reorientation.

DNA:RNA hybrid helixes offer several combinations between A-, B- and C-types helixes in the strands *I* and *II*. Thus for example, the microhelix 5′-UUU-3′ with only uridine in strand *I* (RNA) of A-type and with dA in strand *II* (DNA) of C-type appeared more stable (6.7 kJ/mol, Appendix A) than that with strand *II* of A-type. In this strand *II* the pseudorotation phase angle P was 102.4° with C-type (T10 symmetry) while it was 5.9° with A-type (T23). Although both microhelixes were stable, that with the C-type in strand *II* led to a noticeable increment in the helical parameters, Appendix A. Thus, the helix diameter *d* noticeably increased (17.717 vs. 17.178 Å in A-type of strand *II*), as well as the helical radius R (9.160 vs. 8.603 Å), while the propeller twist θp was remarkably reduced. It led to that H-bonds ① and ② were stronger and thus a higher stability.

#### 3.5.3. Effect on the Helical Parameters of the Mixed Purine–Pyrimidine Stacks Versus Purine–Purine or Pyrimidine–Pyrimidine Stacks 

Comparing the parameters of the mixed 5′-AUA-3′ microhelix versus the 5′-UUU-3′ microhelix, the main differences observed were the following:(i)The microhelix 5′-AUA-3′ was remarkably more stable than 5′-UUU-3′ in both the A-type and B-type, i.e., helixes in which the nucleotides appeared noticeably mixed in each strand were more stable than those in which the mixing was small. It was due to the highest H-bond interactions and stacking interactions between the different planes of nucleotides in mixed strands.(ii)The planarity of the pyrimidine ring in strand *I* was noticeably reduced in the A-type and increased in the B-type as compared to in 5′-UUU-3′. This increment in planarity in the B-type was because of the H-bond ⑩ which deformed the planarity of the uracil ring and weakened the H-bonds ① and ②.(iii)The β and ε′ exocyclic torsional angles in the B-type helix were slightly incremented in ca. 5°, and α and δ slightly decreased by a similar amount ca. 5°, as compared to in 5′-UUU-3′. The differences in these parameters were small in the A-type.(iv)The exocyclic torsional angle ζ was noticeably reduced in B-type, 19°, but only very small, 1°, in the A-type. The highest change in A-type corresponded to the ε torsional angle, 50.2° in 5′-AUA-3′ vs. −160.0° in 5′-UUU-3′. The pseudorotation phase angle P was also noticeably changed, with a change in the symmetry of the furanose ring, from ^2^E to T12 in B-type and from ^3^E to T23 in A-type.(v)The rise parameter Dz between the base pair planes (Scheme 4), as well as the diameter of the helix d and the helical radius R, were noticeably increased in the B-type helix 5′-AUA-3′. This feature was also observed in the A-type but in less quantity, i.e., the regions of the ADN:RNA helix not mixed in each strand would appear more compressed.(vi)The torsional angle N$1_{0}^{U,S}$–C$1\prime_{0}^{U,S}$···C$1\prime_{0}^{A}$–N$9_{0}^{A}$ of the base pair appeared significantly reduced, as well as the propeller twist θp dihedral angle, the inclination parameter INC of the base pair related to the origin of the microhelix axis O_0_, the torsional angle aM_0_O_0_Z between the helical axis through O_0_ and the orthogonal axis to the uracil ring plane n, and the torsional angle cR_0_O_0_Z between the helical axis through O_0_ and the orthogonal axis to the adenine ring plane n.(vii)The dipole moment was remarkably reduced in the A-type, 14.347 D in 5′-UUU-3′ vs. 6.358 D in 5′-AUA-3′, i.e., the regions of the ADN:RNA helix not mixed (with the same nucleoside) in each strand appeared more available for the water molecules to enter inside of the helix, and therefore these regions appeared more easy to interact with pharmacological compounds soluble in water.

#### 3.5.4. Effect of the Sulphur Atom in the Microhelix

Several noticeable effects were observed when the sulphur atom was inserted in the helix, Figure 9, Appendix A. They were the following:(i)A slight increment in the non-planarity of the uracil ring, especially in the torsional angles involving N1 atom in the microhelixes A-type and N3 atom in B-type. The highest effect was observed in the 5′-A s^2,4^U A-3′ microhelix of B-type.(ii)A slight increase of the ^3^E pucker, especially in microhelixes 5′-U s^2^UU-3′ of A-type, while it was decreased with s^4^U. This feature is in accordance to the increment observed in the ^3^E pucker abundance when s^2^U is substituted in the anticodon loop of tRNA [66]. This s^2^U substitution appeared to produce a small increase (2.9 kJ/mol) in the stability of s^2^U:A base pair as compared to U:A in RNA duplexes [7].(iii)A small change of ca. 2° in the exocyclic torsional angles of the microhelix A-type, and a large change in B-type, in especial ca. 8° in ζ with the S2 substitution in 5′-UUU-3′, and ca. 13° with s^2,4^. These changes were smaller when the sulphur atom was inserted into 5′-AUA-3′. However, these changes were not significant for a very flexible helix, because we have found [28] that the highest changes appear with the substitution of uridine by cytidine, such as in the 5′-CUC-3′ microhelix of A-type, with changes of ca. 30° in ζ and ca. 40° in β, as compared to changes in 5′-UUU-3′, and with values slightly lower in the B-type.(iv)With the s^2^U nucleoside, a lengthening of the intermolecular N3–H3···N1^A^H-bond ①, and a small shortening of the O4···H6^A^–N6^A^ H-bond ② was observed in 5′-AUA-3′ of the A-type, and shortening of both H-bonds in 5′-UUU-3′. However, with the s^4^U and s^2,4^U nucleosides both H-bonds were always remarkably lengthened. In microhelixes of B-type the changes with s^2^U were negligible, but similar with s^4^U and s^2,4^U, Table 6.

This lengthening (weakening) of H-bonds ① and ② by the sulphur atom, especially with S4 atom, led to a distortion of the helix. However, with S2 the distortion of the helix was small, but substitution with S2 had the advantage that the possible wobble pairing [66] between S2 of s^2^U nucleoside and the imino proton of guanosine (G), S2···H^G^–N^G^ was weakened, favoring correct binding with adenosine (A). That is to say that the formation of U:G mismatches was reduced as has been reported [66] with s^2^U.
(v)A slight increment of the rise parameter Dz (base stacking) in the 5′-UUU-3′ microhelixes, especially with s^4^U and s^2,4^U nucleotides.(vi)A slight lengthening of the helix diameter d in A-type microhelixes (with the exception of s^4^U) and shortening in B-type, with the exception of s^2^U. The helical radius R always increased with the exception of s^4^ in 5′-AUA-3′ of the B-type. A general slight deformation in the helical parameters INC and θ_p_ was also observed.(vii)A slight increment in the dipole moment (μ) in the B-type, in accordance to a higher stability of the RNA helixes with sulphur was observed in water solution [26,66]. However, in the A-type a trend was not observed, and thus in this case the effect of sulphur could not be considered of special interest because of the very large range of observed values of μ, from 17.4 D in 5′-UUU-3′ to 6.4 D in 5′-AUA-3′.

#### 3.5.5. Hydration of Few Microhelixes

An improvement in the optimization of the microhelix was obtained with the inclusion of one sodium atom H-bonded to each phosphate group and up to 18 water molecules H-bonded mainly to these sodium atoms and to the polar groups of the structure, Figure 10, Appendix A. This simulation was mainly carried out for several selected microhelixes of B-type (C2′-*endo*) because they are the most stable forms [28].

Although the water molecules were mainly inserted close to the sodium atoms and phosphate groups, however other water molecules were also included in the minor and major grooves of the base pairs, according to the hydration of purines and pyrimidines. Thus, they have one main hydration site in the minor groove and one (pyrimidine) or two (purine) in the major groove. The presence of a 2′-OH group in the minor groove of strand *I* attains more polar nature, and the H-bonding of water molecules. High-resolution X-ray study of an RNA duplex [70] has reported the existence of a network of water molecules spanning the minor groove of an RNA A-form helix. The presence of the 2′-OH groups in the primer brings water molecules into the regions that are hydrophobic in the DNA duplex.

The geometry of hydration around many bases in the helix was very similar to that around the isolated bases, but the local deformations of the DNA geometry might cause redistribution of water densities so that the hydration sites might be in different positions from that in the isolated bases. Solvent-oriented H-bond acceptors were often desolvated during duplex formation. The greater the strength of the H-bond acceptor, the greater was the free energy penalty for this process. Removing the H-bond acceptor entirely or substituting it with a weaker acceptor, such as the sulphur atom, was shown to partially alleviate this energy penalty.

The main effects observed in the microhelix 5′-AUA-3′ + 4Na + 8H_2_O of A-type were as follows:(i)A noticeable shortening of H-bond 
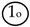
 with the hydration (ca. 0.13 Å), while a slightly lengthening of H-bond 
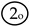
 , Appendix A and Scheme 2.(ii)A slight compression of the microhelix by the water molecules, with a shortening of the P_1_···P_2_ (~0.15 Å) and P_3_···P_4_ (~1 Å) distances, the helix diameter d (~0.4 Å) and the helical radius R (~0.1 Å). The very large shortening in the P_3_···P_4_ distance (strand *II*) was due to the strong H-bond of the water molecule w_1_ (Figure 10), which acts as a bridge between the oxygen atoms of the neighboring phosphate groups, and therefore approaching both phosphate groups. This compression of the helix also led to a noticeable modification in other backbone parameters, Appendix A, especially those in strand *II*.(iii)The compression of the microhelix also affected the rise parameter Dz, Appendix A, with a noticeable decrease in its value, 0.03 Å in Dz_−1_ and 0.24 Å in Dz_0_.(iv)A noteworthy change in the propeller twist θp (~9°) and a small increment in the aM_o_O_o_Z (~2°) and cM_o_O_o_Z (~3°) helical parameters.(v)A small increment of ca. 2° in the furanose pucker P, but it was enough to change its symmetry to ^3^E.(vi)A strong increment of the dipole moment μ, ca. 3 D, by the water molecules.

## 4. Summary and Conclusions

The effect of the sulphur atom on the uracil ring was analyzed as a nucleobase, as a nucleoside and as a nucleotide in the single base pair and in a microhelix. The following conclusions were drawn:The ring in the uracil molecule appeared easier to be deformed than when it was thio-substituted. This high flexibility of the ring in uracil molecule could explain the great biological importance of this molecule, in special for the genetic code. In contrast, a lower flexibility of the uracil ring with thio-substitution could explain the less biological importance of these thio-molecules.The water molecules slightly affected the planarity of the uracil ring, and modified its stability. In the isolated state 2TU was slightly less stable than 4TU, but with the hydration it was the reverse. With the progress of the hydration, the trend in the different geometrical parameters appeared in general similar among uracil and its thio-substituted derivatives.Molecular docking of 2-thiouracil against three different pathogens such as *Bacillus subtilis*, *Escherichia coli* and *Candida albicans* revealed a strong intermolecular H-bonded interaction between the N–H site of ASN33 and O atom of 2TU and with the N–H site of GLU32 and O atom of 2TU. Thus, 2TU could be used to be more effective on gram negative strains.With the sulphur-substitution the H-bonds of the WC pairs appeared somewhat weakened, especially on O4 position, slightly destabilizing the pair, i.e., the pairs with the sulphur atom inserted in the uracil ring were less stable than those in the canonical form.Depending on where the 2′-OH group was H-bonded, four types of microhelixes were optimized. However, here only the A-type and B-type were analyzed in detail because of their higher stability. A-type helixes are characterized by the intra-strand H-bond 2′-OH(n)···O4′(n + 1) with the furanose ring atom O4′, while B-type helixes are described by the H-bond 2′-OH(n)···O7′(n) with the adjacent O7′ oxygen atom of the phosphate group.The A-type microhelix with only uridine in strand *I* (RNA) appeared to modify/prefer the sugar ring orientation of dA in strand *II* (DNA) as T10 (C-type) instead of as ^3^E (A-type). Although both microhelixes were stable, however with C-type in strand *II* was more stable than with A-type, and it led to a lower deformation of the helix. However, with the C-type in strand *I*, the microhelix was noticeably less stable than with A-type.Purine–pyrimidine stacks such as in 5′-AUA-3′ appeared remarkably more stable than purine–purine or pyrimidine–pyrimidine stacks as in 5′-UUU-3′. That is, ADN:RNA not mixed helix regions were more compressed (higher Dz, d, R) and strongly rotated each base in the base-pair (higher θp, INC, aM_0_O_0_Z, cR_0_O_0_Z). In addition, these regions could be more available for water molecules and soluble pharmacological compounds to enter and interact with the helix.The base pairs with sulphur led to a slight increment in the non-planarity of the uracil ring, and to a small change in most of the exocyclic torsional angles. However, the change was noticeably large in the torsional angle ζ of B-type microhelixes. These effects did not appear to be significant in a very flexible helix with very large changes of ca. 40° when uridine was substituted by cytidine.The sulphur atom inserted in 5′-UUU-3′microhelixes, especially in s^4^ and s^2,4^ positions, led to a slight increment of the rise parameter Dz (base stacking), and to a slight lengthening of the helix diameter d in A-type microhelixes and shortening in B-type, with the exception of s^2^U. The helical radius R usually increased with the sulphur substitution. The sulphur atom also affected the helical parameters INC and θp, with a slight modification.The addition of Na atoms and water molecules on the microhelix led to a noticeable shortening of H-bond ① and slight lengthening of H-bond ② in 5′-AUA-3′. It also led to a slight compression of the microhelix with shortening of the P_1_···P_2_ and P_3_···P_4_ distances, the helix diameter d, the helical radius R and the rise parameter Dz. This compression of the microhelix could be only due to a very large modification of all the backbone parameters. This modification also affected the propeller twist θp (with a large change), and other helical parameters. A strong increment of the dipole moment µ of the cluster was also obtained with the hydration, especially in B-type microhelixes.The thio-uridine nucleosides generally had a larger µ than those with uridine. It means that in water solution the bonding of the thio-uridine nucleosides to the primer appeared facilitated over uridine nucleoside. Since the WC pair with the sulphur atom was weaker, it could delay (or deform) the growing of the RNA viral/tumoral, and therefore could explain the use of these thionucleo bases as pharmaceutical drugs.

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
