# Peer review of "Biomolecules of 2-Thiouracil, 4-Thiouracil and 2,4-Dithiouracil: A DFT Study of the Hydration, Molecular Docking and Effect in DNA:RNAMicrohelixes"

_ijms, 2019, doi:10.3390/ijms20143477_

Round 1
Reviewer 1 Report
In my opinion, it is a good paper, clearly written and in good English, reporting on an interesting analysis of the effect of the sulphur atom on the uracil ring as nucleobase, as nucleoside, and as nucleotide in the single base pair and in a microhelix. My recommendation is publishing after some minor changes have been made by the authors. Some tables should be included as supplementary material, for example tables 7, 8, 9 and 10. Similarly, the number of figures seems somewhat excessive.
Author Response
I send it in the enclosed attachment

Reviewer 2 Report
This manuscript account for a computational investigation on the solvation effects of 2-thiouracil, 4-thiouracil and 2,4-dithiouracil systems. Explicit solvent for the first and second shells have been considered. In addition, some docking calculations of 2,4-dithiouracil ligand with various targeted proteins have been also performed. Furthermore, different DNA:RNA hybrid microhelixes with uridine, 2-thiouridine, 4-thiouridine and 23 2,4-dithiouridine nucleosides have been studied. The investigation has been done by using the M062X exchange-correlation functional. The subject is of interest and results consistent with the obtained data. The manuscript is suitable for the publication but the following points must be considered:
- The authors write M062X method. This is not correct since M062X is an exchange-correlation functional in the framework of DF method. Please change along the text;
- The manuscript is difficult to read and follow since the presence of a lot of tables and figures. Many of them (e.g. the structural information) can be shifted in a SI section;
- It is not clear if after the geometry minimization frequencies computations have been done to ensure that the obtained structures are real minima;
The main scientific point concern the selection of the starting structures. I think that a preliminary MD computations should be the best way to select the starting geometries. A comment on this point is welcome
Author Response
I send it in the enclosed attachment

Reviewer 3 Report
The authors have studied the structural changes in uracils through the hydration. This kind of study is very significant for the field concerning with the nucleic acid chemistry. Moreover, the calculation they performed is very sophisticated and of high value from both sides of quality and amount.
On the other hand, the format may not be suitable for the publication. For example, the number of pages is not present correctly. The authors used itemization too much in the manuscript. Is such itemization necessary?
From the academic view, I have some comments as follows:
Figures 4-6: I could not find out the importance of this data. This opinion is based on a question: How did the authors add the number of water molecules? (manually? If so, is there any arbitrariness in numbering the water molecules? -> Such numbering is not meaningful, which leads to the beginning of this comment.)
Table 3: the efficient digit is something wrong. The efficient digit of energy is at most 2. (e.g. H-bond energy x -20.5177 => o -20.52 or -20.5)
Tables 7-9: This is totally my personal opinion, but please delete unnecessary data. (Not all of data seems to be used in the manuscript. Such data should be moved to Supplementary data.)
l.520-523: The representation should be checked. (hard to understand)
Therefore, I recommend this manuscript for the publication after some revisions regarding to my comment mentioned above.
Author Response
I send it in the enclosed attachment
